# Comparison of Adrenal Tumor Size in Ultrasound Examinations with and without the Use of a Contrast Agent

**DOI:** 10.3390/medicina55050165

**Published:** 2019-05-20

**Authors:** Mateusz Winder, Wojciech Spychałowicz, Aleksander J. Owczarek, Jerzy Chudek

**Affiliations:** 1Department of Internal Medicine and Oncological Chemotherapy, School of Medicine in Katowice, Medical University of Silesia in Katowice, ul. Reymonta 8, 40-029 Katowice, Poland; chj@poczta.fm; 2Department of Statistics, Department of Instrumental Analysis, School of Pharmacy with the Division of Laboratory Medicine in Sosnowiec, Medical University of Silesia in Katowice, ul. Ostrogórska 30, 41-200 Sosnowiec, Poland; aowczarek@sum.edu.pl

**Keywords:** adrenal incidentaloma, size, ultrasound, CEUS, contrast, CT

## Abstract

*Background and objectives:* Patients diagnosed with incidentally found adrenal tumors (incidentaloma) that do not meet the criteria for surgical treatment require follow-ups with repeated imaging. The aim of this study is to compare the accuracy of the measurements of the adrenal tumor size in ultrasound (US) with and without contrast in comparison to computed tomography (CT) or magnetic resonance (MRI). Further, this study attempts to answer the question of whether contrast-enhanced ultrasound (CEUS) can improve imaging accuracy and replace CT/MRI in the monitoring of patients with adrenal tumors. *Materials and Methods:* The retrospective analysis included 79 adult patients with adrenal incidentalomas not exceeding a dimension of 6 cm who underwent a CT or MRI scan, US, and CEUS with the use of SonoVue in two-dimensional (2D) and three-dimensional (3D) projections and Doppler techniques. Tumor vascularization in CEUS was classified as follows: peripheral, peripheral-central, central, or poor. *Results:* Of 79 adrenal tumors, 48.1% showed peripheral, 29.1% showed poor, 21.5% showed peripheral-central, and only 1.3% showed central vascularization. The median volume of tumors detected with CEUS (69.9 cm^3^) was significantly higher than with US (44.5 cm^3^) and CT or MRI (57.1 cm^3^). The relative error of the adrenal volume with CEUS compared with CT or MRI was significantly higher than with standard US, regardless of the type of tumor vascularization. *Conclusions:* CEUS does not improve the accuracy of adrenal tumor size assessment regardless of the type of vascularization.

## 1. Introduction

Patients with incidentally diagnosed adrenal tumors (incidentaloma) frequently do not meet the criteria for surgical treatment and require periodic controls with repeated imaging. When the diameter of the adrenal tumor is ≤3 cm, and the image corresponds to a typical lipid-rich adenoma, it is recommended that imaging examinations are performed every 12 months for five years. In the case of larger tumors having a less characteristic image, in the first year of observation, follow-up examinations should be considered every 3–6 months and then once a year. A rapid enlargement of the tumor, its diameter exceeding 5 cm, and the appearance of malignancy features of the tumor are indications that surgical treatment is required [1].

The recommended examination method in the imaging of newly diagnosed adrenal tumors is computed tomography (CT) with contrast administration or magnetic resonance (MRI). However, in accordance with the recommendation of the PTE (Polish Endocrine Society), both ultrasound (US) and CT without contrast may be used to monitor the size of adrenal tumors [1]. Under optimal conditions, normal adrenal glands can be visualized by US from abdominal access with a convex transducer when their size is 5–10 mm [2]. In adults, conditions frequently differ from optimal, hence focal lesions smaller than 20 mm in the left adrenal gland may be invisible in US [2]. In the case of adrenal tumors, a standard US examination carries a high risk of false positive and negative results. Therefore, US scanning is recommended in the monitoring of adrenal lesions visible in this examination, with a diameter >3 cm, especially localized on the right side of the body [1].

The applicability of contrast-enhanced ultrasound (CEUS) imaging in the diagnosis of adrenal gland lesions has been the subject of few studies, which were limited to attempts to assess the character of tumors based on their vascularity. It was shown that the analysis of characteristic patterns of adrenal tumor vasculature in CEUS may increase the sensitivity of detecting malignant lesions up to 100% with a specificity of 67%–82% [3,4]. Other reports assessed these values much more cautiously, i.e., at the level of 50% and 68%, respectively [5].

Publications comparing the size of tumors estimated with the employment of US, CEUS, and CT or MRI are mainly related to lesions in the liver and pancreas. The authors of one of them combined images of hepatocellular carcinoma and metastatic changes in the liver measured with CEUS and CT or MRI and demonstrated that there were no significant differences in the obtained measurements [6]. However, in a study comparing the dimensions of pancreatic tumors, a significant relationship between CEUS and CT dimensions was found, wherein the dimensions measured with CEUS were smaller than those coming from CT [7]. Moreover, it has been shown that the sensitivity of CEUS examinations is greater than that of MRI in the case of the detection of pancreatic adenocarcinoma when its dimension does not exceed 3 cm [8].

This study’s aim is to compare the accuracy of adrenal tumor size measurements with the employment of US scanning with and without contrast and with CT or MRI and to attempt to answer the question of whether CEUS examinations can improve the imaging accuracy and replace CT in the monitoring of patients with adrenal tumors.

## 2. Materials and Methods

This retrospective study comprised 85 patients (71 women and 14 men) aged 30–81 (mean age 61 ± 16 years), who were hospitalized at the Department of Internal Medicine and Oncological Chemotherapy due to incidentally detected adrenal tumors in 2010 and 2011. Each patient underwent CT or MRI scans and US examinations before and after intravenous administration of SonoVue contrast (Bracco International B.V., Amsterdam, Netherlands) in two-dimensional (2D) and three-dimensional (3D) projections using Doppler techniques, as seen in Figure 1 and Figure 2. CT/MRI and US/CEUS examinations were performed at intervals of up to 6 weeks. US scanning was performed by a certified US specialist using a Siemens Acuson Antares instrument (Siemens Medical Solutions USA, Inc. Mountain View, CA 94043, USA) with a 2–6 MHz convex transducer. Based on the performed imaging, hormonal tests, and pathological evaluation of oligobiopsy, 81 incidentalomas, 2 pheochromocytomas, 1 case of adrenocortical carcinoma, and 1 liver cyst were diagnosed.

In the final analysis, 79 patients were included, and 6 cases with incomplete data and in which the maximum tumor size exceeded 6 cm were excluded. 

CEUS was a part of the routine assessment introduced to the patient work-up and was approved by the institution’s supervisors. The study protocol was accepted by the Bioethics Committee on 15 December 2009 (KNW/0022/KB1/153/I/09). All patients participating in the study gave written informed consent for the performed examinations and procedures. Figures showing tumors were also obtained through the patients’ informed consent for their use.

### 2.1. Data Analysis

Based on the US, CEUS, and CT or MRI measurements of adrenal gland tumors, we calculated their volumes, which were subsequently employed in a further comparative assessment. The types of adrenal tumor vascularization in our study were classified as follows: 1. lack of vascularization, 2. peripheral vascularization, 3. peripheral-central vascularization, and 4. central vascularization.

### 2.2. Statistical Analysis

Statistical analysis was performed using STATISTICA 10.0 PL (Tibco Software Inc. Palo Algo, CA, USA) and StataSE 12.0 (StataCorp LP, College Station, TX, USA). Cases with missing data were excluded from the analysis. No data imputation was done. The volumes of adrenal gland tumors were calculated either as a volume of ellipsoid V = 4/3∙π∙a∙b∙c, where a, b, and c are dimensions in CT/MRI or as a V = 4/3∙π∙a∙b^2^, where a and b are dimensions in US and CEUS (a—anterior-posterior, b—coronal, and c—transverse). Nominal and ordinal data were expressed as percentages, while interval data were expressed as a mean value ± standard deviation in the case of normal distribution or as median (lower quartile – upper quartile) in the event of data with skewed or non-normal distribution. Distribution of variables was evaluated by the Shapiro–Wilk test and quantile–quantile plot, and homogeneity of the variances was assessed by the Levene test. Adrenal volume in the US, CEUS, and CT/MRI groups was compared with one-way analysis of variances (ANOVA) in cases of normal data distribution or after logarithmic normalization in cases of skewed data (if appropriate) with the Tukey’s post-hoc test. Comparisons of adrenal volume between two classes of tumor vascularization were done with the t-Student test for independent data (for original data or after logarithmic normalization in cases of skewed data). The agreement between US, CEUS, and CT/MRI groups was measured with Lin’s concordance correlation coefficient. The results of the comparison between the US, CEUS, and CT/MRI groups were presented with bagplots (Figure 3 and Figure 4). The inner polygon, constructed on the basis of Tukey depth, contains at most 50% of the data points. The observations that are not marked as outliers are surrounded by a loop, the convex hull of the observations within the fence. Observations outside the fence are flagged as outliers and shown with the asterisk symbol (*). The rectangle near the center of the graph shows the depth median, the point with the highest possible Tukey depth. The ellipse around the convex hull shows 95% confidence interval ellipse. The thin dashed line shows a reference line (y = x), and the thick line shows the orthogonal regression function. Assessment of the relationship between variables was done with orthogonal least square regression, and correlation between variables was measured with the Pearson linear correlation coefficient, adhering to appropriate requirements. The relative error was calculated as a fraction of the difference between US or CEUS and CT/MRI measurements. Statistical significance was set at a p value below 0.05. All tests were two-tailed.

## 3. Results

Out of the 79 adrenal tumors, 38 (48.1%) had peripheral vascularization (type II), 23 (29.1%) pointed to the lack of vascularization (type I), 17 (21.5%) had peripheral-central vascularization (type III), and only 1 (1.3%) showed central vascularization (type IV) in CEUS. The two pheochromocytomas showed peripheral vascularization and strong peripheral-central vascularization, respectively. The only pathologically confirmed adrenal carcinoma showed strong peripheral-central vascularization.

In the case of adrenal tumors classified in CT/MRI scanning as adrenal adenomas, the dominant vascular pattern was peripheral, observed in 37 cases (47.3%). However, this was not a characteristic feature due to the large number of tumors classified in CT/MRI as adenomas that did not show vascularity (*N* = 23, 30.7%) or that exhibited peripheral-central vascularization (*N* = 15, 20%). Only one adrenal adenoma showed central vascularization.

### Adrenal Tumor Volume Measurements with Various Methods

The statistical analysis showed that there are significant differences in the volumes of adrenal tumors calculated on the basis of the measurements obtained in US, CEUS, and CT/MRI. It was found that the mean volume of tumors in CEUS scanning was significantly higher than the mean volume in US and CT/MRI examinations (*p* < 0.001). However, there were no significant differences between adrenal volume measurements in non-contrast ultrasound and CT/MRI (*p* = 0.99) as can be seen in Figure 5 and Table 1. Regardless of the noticed differences, a very strong correlation of log_10_ adrenal volume was found between US and CT/MRI (r = 0.84; *p* < 0.001) as well as between CEUS and CT/MRI (r = 0.82; *p* < 0.001) as can be seen in Figure 3 and Figure 4. The Lin’s concordance coefficient between US and CT/MRI measurements was 0.840 (95% CI: 0.774–0.905; *p* < 0.001) and that between CEUS and CT/MRI measurements was 0.796 (95% CI: 0.718–0.875; *p* < 0.001).

The calculations of the relative adrenal volume measurement error between US and CEUS compared with the volumes obtained with CT/MRI scanning showed that CEUS examination is subject to a higher probability of making a measurement error [38.9% (95% CI: −33.7 to 121] for lesions with peripheral or peripheral-central vascularization and 7.23% (95% CI: −12.7 to 64.8) for poor or central vascularization) than a standard ultrasound examination (*p* <0.01), as can be seen in Table 1 and Figure 6.

Peripheral or central-peripheral types of adrenal vascularization affected neither the median adrenocortical volume in each of the measurement methods nor the relative adrenal volume measurement errors.

## 4. Discussion

The results of our study point to the risk of overestimating the adrenal tumor volume in CEUS in comparison to CT/MRI examinations, especially in tumors with poor peripheral vascularization. This may result from glare artifacts caused by excessive signal amplification or bright contrast enhancement in the near-field that attenuate visualization of distally located lesions in adrenal glands that are not clearly isolated by poor peripheral vascularization [9].

It should be emphasized that in the available literature, there are very few reports on the diagnosis of adrenal tumors using the CEUS method. The previous reports were very optimistic and highlighted the high sensitivity of CEUS examination in determining the character of adrenal focal lesions [3,4]. The authors of those reports linked malignancy with hypervascularization, early arterial enhancement signal, and tumor size exceeding 4 cm. Our analysis of adrenal tumor vasculature in CEUS suggests that about one third of benign adenomas, described in the CT/MRI study, show poor central vascularization (lack of vascularization in 29.1% and central in 1.3%). The measurements of the size of such tumors involve the risk of its overestimation, which may provoke patients’ anxiety and eventually result in their referral for unnecessary surgery. In addition, the concordance correlation coefficient between CEUS and CT/MRI measurements was only 0.796 (95% CI: 0.718–0.875), which precludes the method for consideration as a tool for clinical practice, replacing the current gold standard [10].

Adrenal incidentalomas are not limited to hormonally inactive lesions. Characterization of vascular patterns of other adrenal tumors in CEUS, e.g., pheochromocytoma and cancer, is beyond the scope of this study due to the small number of other tumors. Furthermore, the presented study mainly included tumors with a volume exceeding 4 cm^3^ (of medium or large size). For this reason, this study is not suitable for determining the importance of the CEUS method in imaging small adrenal tumors, which should be considered as an important study limitation. The strength of our study rests on relating the obtained measurements in US and CEUS to the reference methods (CT/MRI) for adrenal tumor imaging.

## 5. Conclusions

In conclusion, CEUS does not improve the accuracy of adrenal tumor size assessment regardless of the type of vascularization. Therefore, this study does not indicate a broad applicability of the CEUS method in the monitoring of adrenal tumor growth.

## Figures and Tables

**Figure 1 medicina-55-00165-f001:**
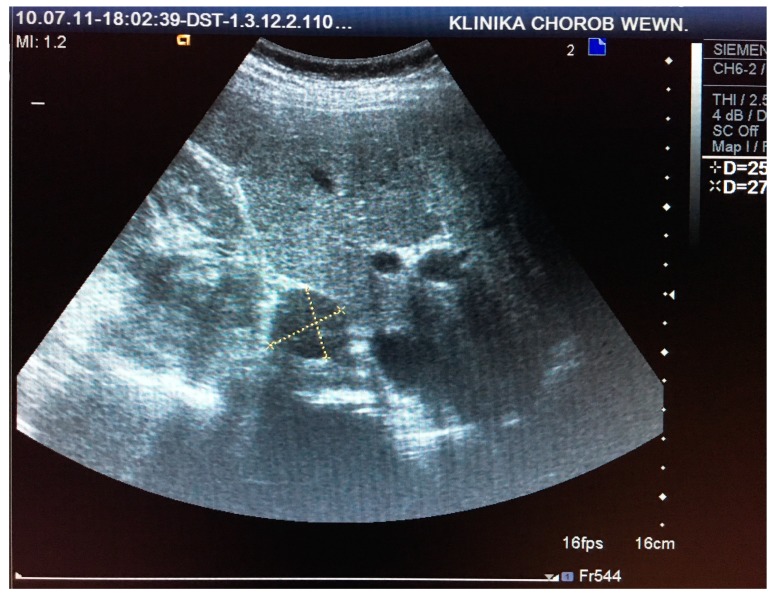
Visualization and measurement of adrenal tumor in two-dimensional (2D) ultrasound (US) examination.

**Figure 2 medicina-55-00165-f002:**
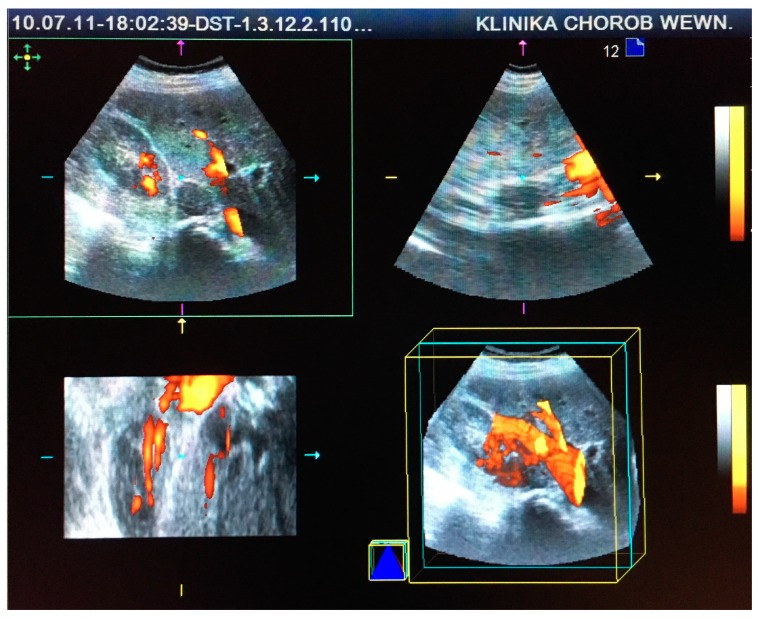
Visualization and vascularization analysis of the same adrenal tumor in contrast-enhanced ultrasound (CEUS) using power Doppler and three-dimensional (3D) techniques.

**Figure 3 medicina-55-00165-f003:**
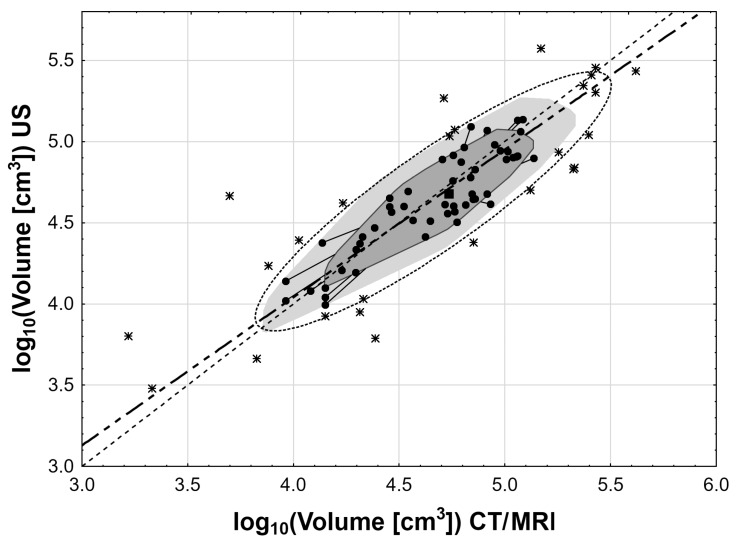
The bagplots for the comparison of log_10_ of adrenal volumes between the US group versus the CT/MRI group. The observations that are not outliers are marked with dots, while outliers are shown with the asterisk symbol. The rectangle near the center of the graph shows the depth median.

**Figure 4 medicina-55-00165-f004:**
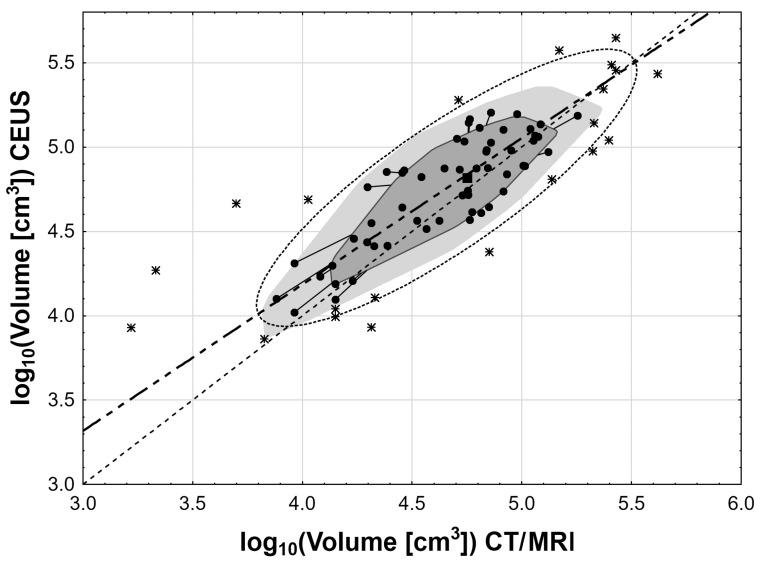
The bagplots for the comparison of log_10_ of adrenal volumes between the CEUS group versus the CT/MRI group. The observations that are not outliers are marked with dots, while outliers are shown with the asterisk symbol. The rectangle near the center of the graph shows the depth median.

**Figure 5 medicina-55-00165-f005:**
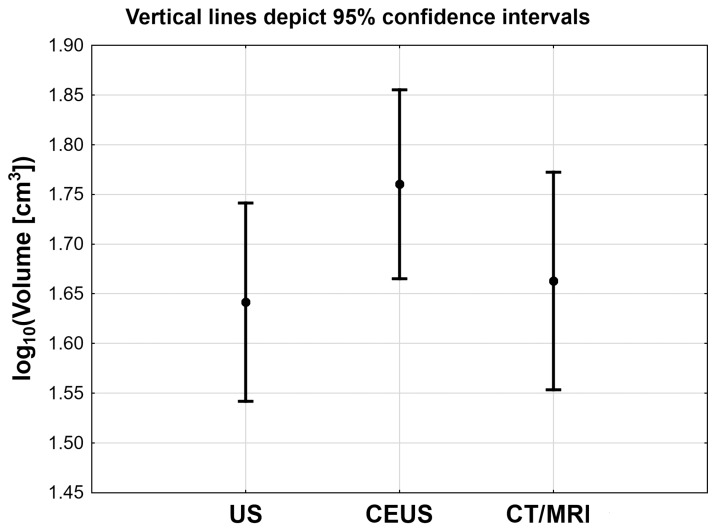
Mean adrenal tumor volume with 95% confidence interval based on measurements with US, CEUS, and computed tomography (CT) or magnetic resonance (MRI).

**Figure 6 medicina-55-00165-f006:**
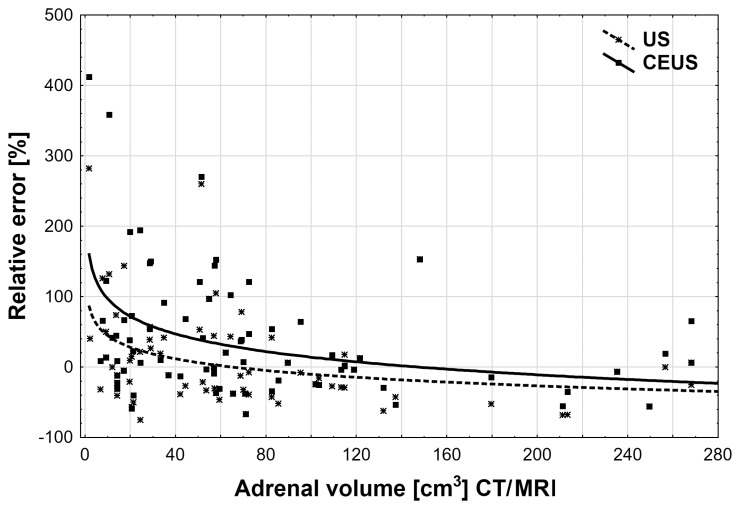
The relative errors (%) of adrenal volume measurements between the US and CEUS groups versus the CT/MRI group, according to adrenal volume measurements in the CT/MRI group (extreme values with error >500% were rejected). Data relationships were fitted with the log10 function: US: 264.1 − 54.8 × log_10_(x); CEUS: 428.7 − 82.9 × log_10_(x).

**Table 1 medicina-55-00165-t001:** Analysis of volume estimation discrepancies by US, CEUS, and CT or MRI in relation to the vascularization class.

	All Cases (*N* = 79)	Vascularization Class		*p*
		None or Central (*N* = 24)	Peripheral or Peripheral-Central (*N* = 55)	
Tumor volume US (cm^3^)	44.5 (23.9–85.9)	39.6 (15.6–77.7)	50.1 (25.9–87.7)	0.12
Tumor volume CEUS (cm^3^)	69.9 (28.7–115.0)	48.7 (23.9–96.0)	74.9 (36.7–112.7)	0.20
Tumor volume CT/MRI (cm^3^)	57.1 (20.6–101.8)	34.8 (19.7–65.3)	62.0 (24.4–114.6)	0.12
US error (%)	−7.6 (−34.4–39.0)	13.6 (−40.5–43.3)	−11.1 (−31.9–20.7)	0.56
CEUS error (%)	11.5 (−14.4–68.5)	38.9 (−33.7–121)	7.23 (−12.7–64.8)	0.41

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
