# Peer review of "Comparison of Adrenal Tumor Size in Ultrasound Examinations with and without the Use of a Contrast Agent"

_medicina, 2019, doi:10.3390/medicina55050165_

Round 1
Reviewer 1 Report
I found that the article by Winder et al. has an interesting development. The aim of the work is to compare the accuracy of the measurements of the adrenal tumor size in ultrasonography without (US) and with contrast (CEUS) in comparison to computed tomography (CT) or magnetic resonance (MRI), and check if CEUS can improve the imaging accuracy and replace the CT/MRI. They analyzed 79 patients and, for patients with tumors with volume bigger than 4cm3, CEUS did not improve the accuracy. Even tough the result of their analyze been negative, it is an important contribution.
I
have the following comments:
1)
Each Figure should have its own caption. Please describe what you
want the readers to know in Figures 1 and 2. Tell which imaging
method represents each figure. Even if the reader is not familiar
with the imaging techniques, the caption should help him.
2)
In page 4, line 106, which dimensions are a, b and c in CT/MRI and
which are a and b in US and CEUS?
3)
In Figure 4 and 5, change the labels to match the figure, for example
“Figure 4. Bagplots for the comparison of log10 of adrenal volumes
between US group versus CT / MRI group.”
4)
During the paper you use MRI and for the figures labels you use NMR.
Please replace the NMR for MRI.
Finally,
I suggest publication after revisions.
Author Response
Dear Sir / Madam,
Thank you for your comments.
I have already made the necessary changes in all figures and their captions:
For Figure1 "Visualization and measurement of adrenal tumor in 2D US examination."
For Figure 2 "Visualization and vascularization analysis of the same adrenal tumor in CEUS using power Doppler and 3D techniques."
Concerning the dimensions by a, b and c we ment the anterior-posterior, coronal or craniocaudal and transverse measurements. I also clarified this in the article.
Kind regards,
Matt Winder
Reviewer 2 Report
This study sought to compare accuracy of adrenal lesion size among various imaging modalities. The key question if contrast enhanced US could improve and replace CT was analyzed. The study concluded that CEUS does not improve accuracy and would not replace current imaging standard. The analysis is to the point to address the question. I do not have any other critical input.
Author Response
Dear Sir / Madam,
Thank you for your review and comment.
Kind regards,
Matt Winder
Reviewer 3 Report
Dear Authors,
The topic of this manuscript is interesting and the main findings are well-presented.
The section of statistical analysis is thoroughly elucidated.
My comments and suggestions for changes are listed below.
NB.
Suggestions for linguistic formulations or changes are highlighted in red.Suggestions for text to be deleted are marked with strikethrough.
Abstract
1. Page 1, line 16
Consider writing in present tense
Consider dividing the sentence into two
Consider moving the abbreviation forward
The aim of the study was is to compare the accuracy of the measurements of the adrenal tumor size in ultrasonography (US) without contrast (US) and with contrast (CEUS) in comparison to computed tomography (CT) or magnetic resonance (MRI), . Further, the study and an attempts to answer the question whether contrast-enhanced ultrasonography (CEUS) can improve the imaging accuracy and replace the CT / MRI in the monitoring of patients with adrenal tumors.
2. Page 1, line 20-21
Consider to delete the parentheses
The retrospective analysis included 79 adult patients with adrenal incidentalomas (not exceeding dimension of 6 cm)…
3. Page 1, line 25-26
Consider to replace “greater” with higher
The median volume of tumors in CEUS (69.9 cm3) was significantly greater higher than…
4. Page 1, line 26-27
Consider to rephrase if it makes sense
The relative error of the adrenal volume with CEUS was significantly higher than with standard US compared to CT or MRI was significantly higher than with standard US…
Introduction
5. Page 1, line 33-35
Consider to delete the commas
Patients with incidentally diagnosed adrenal tumors (incidentaloma) frequently do not meet the criteria for surgical treatment, and require periodic controls with repeated imaging. When the diameter of the adrenal tumor is ≤ 3 cm, and the…
6. Page 1, line 41
Consider to delete the commas
Recommended examination, in the imaging of newly diagnosed adrenal tumors, is computed…
7. Page 1, line 43
Please delete “ultrasound” as you have already writing the abbreviation in full in the abstract Consider to be consequent and write either ultrasonography or ultrasound in the manuscript
…both ultrasound (US)…
8. Page 2, line 43-44
Consider changing word order and then delete a comma
Under optimal conditions Nnormal adrenal glands, under optimal conditions…
9. Page 2, line 45
Consider to just write the abbreviation US instead of the word ultrasonography
Consider to replace “in” with “by”
… can be visualized in ultrasonography by US…
10. Page 2, line 48
Consider to replace ultrasound with the abbreviation US
…a standard ultrasound US examination…
11. Page 2, line 49
The same as above
Therefore, ultrasound US scanning…
12. Page 2, line 52
Consider writing just the abbreviation as it has been write in full in the abstract
The applicability of contrast-enhanced ultrasound (CEUS) CEUS…
13. Page 2, line 68
…ultrasound US scanning…
14. Page 2, line 69
…whether contrast-enhanced ultrasound CEUS examinations…
15. Page 2, line 70
Consider to delete “the”
…replace the CT…
Materials and Methods
16. Page 2, line 75
…ultrasound US examination…
17. Page 2, line 78
Replace “ultrasound” with “US”
The ultrasound US scanning…
18. Page 2, line 80
Replace “ultrasound” with “US”
…certified ultrasound US specialist…
19. Page 3, Figure 1 and 2
Consider to write the abbreviation instead of the words in full and please be
consequent and write “contrast-enhanced ultrasonography” if you do not want to use the abbreviations
For example you could write the abbreviations in full under the Figure.
Figure 1. Analysis of volume estimation discrepancies by ultrasound (US) and contrast-enhanced sonography (CEUS), and computed tomography (CT) or magnetic resonance imaging (MRI) in relation to the vascularization class.
20. Page 3, line 91
As an example, this comma is probably unnecessary
In the final analysis, 79 patients…
Results
21. Page 7, line 171-173
Consider to only write the abbreviations or to write the abbreviations in full under the Table
Table 1. Analysis of volume estimation discrepancies by ultrasound (US) US and contrast-enhanced sonography (CEUS), CEUS…
In general, to make the manuscript more understandable I suggest:
- Consider deleting some commas
- Consider to be consequent about whether you are using “ultrasound” or “ultrasonography”
- Consider deleting every word of either ultrasound or ultrasonography, after the first time you have mentioned the abbreviation “US”
Author Response
Dear Sir / Madam,
Thank you for your review and comments. It is very helpful.
I have already made the necessary corrections in the article. All words referring to following abbreviations and inappropriately placed commas have been deleted. Sentences have been corrected.
Kind regards,
Matt Winder